# Pregnancy in Women with Arrhythmogenic Left Ventricular Cardiomyopathy

**DOI:** 10.3390/jcm11226735

**Published:** 2022-11-14

**Authors:** Riccardo Bariani, Maria Bueno Marinas, Ilaria Rigato, Paola Veronese, Rudy Celeghin, Alberto Cipriani, Marco Cason, Valeria Pergola, Giulia Mattesi, Petra Deola, Alessandro Zorzi, Giuseppe Limongelli, Sabino Iliceto, Domenico Corrado, Cristina Basso, Kalliopi Pilichou, Barbara Bauce

**Affiliations:** 1Department of Cardiac, Thoracic, Vascular Sciences and Public Health, University of Padua, 35122 Padua, Italy; 2Azienda Ospedaliera di Padova, Via Giustiniani, 2, 35128 Padova, Italy; 3Department of Translational Sciences, University della Campania “Luigi Vanvitelli”, 80138 Naples, Italy

**Keywords:** arrhythmogenic left ventricular cardiomyopathy, pregnancy, desmoplakin, filamin C, plakofillin-2

## Abstract

Background: In the last few years, a phenotypic variant of arrhythmogenic cardiomyopathy (ACM) labeled arrhythmogenic left ventricular cardiomyopathy (ALVC) has been defined and researched. This type of cardiomyopathy is characterized by a predominant left ventricular (LV) involvement with no or minor right ventricular (RV) abnormalities. Data on the specific risk and management of pregnancy in women affected by ALVC are, thus far, not available. We have sought to characterize pregnancy course and outcomes in women affected by ALVC through the evaluation of a series of childbearing patients. Methods: A series of consecutive female ALVC patients were analyzed in a cross-sectional, retrospective study. Study protocol included 12-lead ECG assessments, 24-h Holter ECG evaluations, 2D-echocardiogram tests, cardiac magnetic resonance assessments, and genetic analysis. Furthermore, the long-term disease course of childbearing patients was compared with a group of nulliparous ALVC women. Results: A total of 35 patients (mean age 45 ± 9 years, 51% probands) were analyzed. Sixteen women (46%) reported a pregnancy, for a total of 27 singleton viable pregnancies (mean age at first childbirth 30 ± 9 years). Before pregnancy, all patients were in the NYHA class I and none of the patients reported a previous heart failure (HF) episode. No significant differences were found between childbearing and nulliparous women regarding ECG features, LV dimensions, function, and extent of late enhancement. Overall, 7 patients (20%, 4 belonging to the childbearing group) experienced a sustained ventricular tachycardia and 2 (6%)—one for each group—showed heart failure (HF) episodes. The analysis of arrhythmia-free survival patients did not show significant differences between childbearing and nulliparous women. Conclusions: In a cohort of ALVC patients without previous episodes of HF, pregnancy was well tolerated, with no significant influence on disease progression and degree of electrical instability. Further studies on a larger cohort of women with different degrees of disease extent and genetic background are needed in order to achieve a more comprehensive knowledge regarding the outcome of pregnancy in ALVC patients.

## 1. Introduction

Arrhythmogenic cardiomyopathy (ACM) is an inherited cardiac disease that is characterized by myocardial necrosis and fibro-fatty replacement that predisposes patients to ventricular arrhythmias (VAs), which can even lead to sudden cardiac death (SCD) [1].

Different from the original descriptions—which considered ACM a disease of the right ventricle (RV) with left ventricular (LV) involvement, usually mild when relevant mainly due to a disease progression in association with an advanced RV disease—in the last years it has become evident that LV involvement can be present in early stages of the disease, independently or concurrently with RV involvement [2,3].

Furthermore, recently a left dominant variant of the disease (arrhythmogenic left ventricular cardiomyopathy: ALVC) has been described [3,4]. In this form, the diagnosis can be challenging and it is usually made on the basis of ECG features (inferolateral T-wave inversion and low QRS voltages); VAs of LV origin; prominent LV dilatation/dysfunction in the setting of relatively mild or absent right-sided disease; and in the presence of subepicardial or ring-like late gadolinium enhancement (LGE) following a cardiac magnetic resonance (CMR) assessment [4,5,6]. Genotype–phenotype correlation studies demonstrated that desmoplakin (DSP) and filamin C (FLNC) are the most common disease genes in ALVC [7]. However, many cases are still gene elusive [5,8].

Recent studies on pregnancy in women affected by ACM demonstrated a low rate of major cardiac events during pregnancy [9]. Nonetheless, data on the specific risk and management of pregnancy in women affected by ALVC are not yet available.

In this study, we sought to characterize pregnancy course and outcome in women affected by ALVC through a retrospective evaluation of a series of childbearing patients. Moreover, in order to better characterize the effect of pregnancy in this population we compared the clinical and instrumental data of these patients to those of a group of nulliparous ALVC women.

## 2. Materials and Methods

### 2.1. Study Population

From the entire cohort of probands and family members followed at the Cardiomyopathy Unit of the University of Padua, from 1990 to 2020, we selected a consecutive cohort of 35 female patients diagnosed with ALVC. All patients provided written informed consent before inclusion in the study, in accordance with the protocol approved by the local ethics committee. All clinical investigations were conducted according to the principles expressed in the Declaration of Helsinki. Inclusion criteria for the diagnosis of ALVC were as follows: 1. The presence of a subepicardial LGE pattern with non-ischemic distribution and fatty infiltration at CMR assessment affecting exclusively or predominantly the LV, plus one of the following diagnostic features: A. positive genetic testing for likely pathogenic (LP, class IV)/pathogenic (P, class V) variants associated with ACM; B. presence of family history of ACM, ALVC, or dilated cardiomyopathy (DCM) and/or family history of SCD with autoptic findings in keeping with an ACM form. Moreover, three women belonging to two ALVC families showing LV dimensional, kinetic abnormalities, and carrying DSP P/LP variants who had already received an ICD without a previous CMR evaluation were enrolled.

The study protocol included familial and personal anamneses, 12-lead ECGs, two-dimensional Doppler echocardiograms, 24-h Holter ECGs, CMR assessments, and genetic tests. A comparison of the phenotypic expression and degree of electrical instability of childbearing and nulliparous patients was also performed.

### 2.2. Twelve-Lead Electrocardiograms

Twelve-lead ECGs were performed on a standard speed paper (25 mm/s, 10 mm/mV, and 0.05–150 Hz) and the following parameters were considered: duration of PQ interval, mean QRS duration, right bundle branch block (RBBB—incomplete or complete), left anterior fascicular block, complete left bundle branch block (LBBB), ST-segment alteration (ST elevation > 1.5–2 mm), pathological Q wave, T-wave inversion, and QRS voltages in both precordial and peripheral leads (low voltages were defined when QRS was <5 mm in peripheral leads or <10 mm in precordial leads).

### 2.3. Ventricular Arrhythmias and Heart Failure

Recorded VAs were classified in ventricular fibrillation (VF); sustained ventricular tachycardia (defined as a tachycardia that lasted >30 s) (sVT); (3) non-sustained ventricular tachycardia (defined as three or more consecutive ventricular beats, lasting <30 s, at a rate >120 beats/min) (NSVT); and (4) premature ventricular beats (PVBs). Heart failure (HF) was considered in the presence of the signs and symptoms requiring hospitalization or outpatient clinic evaluation.

### 2.4. Two-Dimensional and Doppler Echocardiography

Two-dimensional echocardiograms were performed with a commercially available Hewlett Packard model 5500 and GE S6 ultrasound machine equipped with a M5S probe. Parasternal, apical, and subcostal views were obtained. In addition, LV function, LV end diastolic volume, RV area, RV function were calculated on an apical four chambers view. Echocardiographic measurements were evaluated according to international recommendations [10]. 

### 2.5. Cardiac Magnetic Resonance

CMR assessment was performed via a 1.5-T scanner (Magnetom Avanto, Siemens Medical Solutions, Erlangen, Germany). All patients underwent a study protocol for myocarditis, including balanced steady-state free precession sequence cine images for morpho-functional evaluation, triple inversion recovery sequences for the detection of myocardial edema, and two-dimensional segmented breath-held fast low-angle shot inversion recovery sequences within 3 min after the administration of intravenous contrast agent (gadobenate dimeglumine; 0.2 mmol/kg of body weight) for the purposes of detecting early gadolinium enhancement (EGE) and 10–15 min for late gadolinium enhancement (LGE). Additionally, we used T1-weighted turbo spin-echo sequences for the purposes of detecting myocardial fat infiltration. The technical details of the CMR sequences and image post-processing analyses have been previously reported [11,12].

### 2.6. Genetic Analysis 

Genetic testing was carried out as previously described [13,14].

### 2.7. Childbearing Group Evaluation 

Among the 35 ALVC patients, 16 (45%) were childbearing and the obstetric courses were assessed through analyses of medical records. For each pregnancy’s gestational duration, type of delivery, birth weight, obstetric complications, and perinatal health were evaluated.

### 2.8. Statistical Analysis 

Data were presented as mean ± standard deviation, median with range, or frequencies with percentages, as appropriate. The normality of the quantitative variables was evaluated through the Shapiro–Wilk test, while their comparison was conducted through the application of the Student’s *t*-test or the Mann–Whitney test, when appropriate. Categorical variables were compared by the chi-square test and Fisher’s exact test, when appropriate. Kaplan–Meier curves were constructed, and a log-rank test was performed in order to assess cumulative lifetime sVT-free survival. The association between pregnancy and major VA (MVA) was tested using Cox regression analysis. The statistical significance for all tests was set for probability values *p* < 0.05. Statistical analyses were performed using SPSS version 27 software for MAC (SPSS, Inc., Chicago, IL, USA).

## 3. Results

A total of 35 women (age at evaluation 45 ± 9 years) were included in the study, of which 18 (51%) were probands. In 18 (51%) a family history of SCD was determined and in 11 (31%) instances of HF were present. The reasons for first evaluation were detection of VAs (*n* = 18, 51%), family history of cardiomyopathy or SCD (*n* = 13, 37%) and chest pain episodes with cardiac enzyme release (*n* = 4, 11%) (Table 1). Regarding medical therapy, 14 (88%) patients used beta-blockers (mainly metoprolol), 7 (50%) patients utilized ACE inhibitors, 1 patient used amiodarone (6%), and 1 patient utilized sotalol with ACE inhibitors (6%). Among the 16 childbearing women, 9 were diagnosed with ALVC after the last pregnancy. In the 7 patients diagnosed before pregnancy, therapy was modified by discontinuing the ACE inhibitor, while beta-blocker therapy was maintained.

### 3.1. Genetic Data

Genetic screening was performed in 34 cases (97%) and genetic variants were identified in 29 (83%). In more detail, 21 (62%) patients carried a DSP, 5 (15%) a FLNC, 2 (6%) a PKP2, and one possessed a DSG2 (3%) genetic variant (Table 2). Moreover, in 5 cases (15%) the genetic test was negative.

### 3.2. Electrocardiographic Findings

ECG assessment showed abnormal features in 25 patients (71%). The most common findings were low QRS voltages in limb (*n* = 23, 65%) and precordial leads (*n* = 11, 31%), followed by negative T waves in left precordial leads (20%). All data on the ECG findings are reported in Table 1. In Figure 1, panel A, the ECG evaluation of a patient with ALVC is present, where low QRS voltages can be observed in the peripheral leads.

### 3.3. Echocardiographic Features

2D-echocardiogram was performed on all subjects. Data on their ventricular functions and dimensions are available in Table 3. LV end diastolic volume (EDVi) was increased in 24 patients (89%) and LV ejection fraction (LV-EF) was reduced in 18 (51%), with the presence of regional kinetic abnormalities in 15 (43%). RV was dilated in 2 patients (6%) and RV systolic function was within limits in all cases.

### 3.4. CMR Findings

CMR assessment was performed on 32 patients (91%). The data on their biventricular dimensions, functions, wall motion alterations (WMAs), and tissue characterization are reported in Table 3.

In the overall population, EDVi was increased in 20 (63%) patients, while LV-EF was reduced in 24 (75%). In 17 cases (49%) LV-WMA was present, mainly involving the infero-posterior wall. RV dimensions and function were within limits in 28 patients (87%), while 4 (12%) showed a mildly increased RV-EDVi, whereas 1 (3%) possessed a mild systolic dysfunction. LV-EF was below 35% in 2 patients, 1 for each group. LV-LGE was detected in all patients with the most common pattern of distribution being the subepicardial stria, mostly located in the basal segments of the inferolateral wall (see Table 3). An example is shown in Figure 1, the description of which is in the caption.

### 3.5. Outcomes of ALVC Pregnancies

Overall, 16 patients experienced 27 singleton viable pregnancies (range 1–3 and mean age at first pregnancy 30 ± 3.9). All patients were on NYHA I functional class before pregnancy and none had a history of HF before pregnancy. During pregnancy, 3 patients (19%) complained of palpitations.

All viable pregnancies resulted in live-born children. Twenty-five (93%) were delivered full term, and two preterm at 37 and 38 weeks. A total of four cesarean sections (25%) were performed: two for ALVC-related reasons and two for primarily obstetric indications (i.e., fetal growth delay, who were both in therapy with beta-blockers). No major obstetric complications occurred during pregnancy. Patients diagnosed with ALVC before pregnancy underwent a close cardiac and obstetrical monitoring. From the cardiological point of view, each patient underwent a cardiological examination including an echocardiographic examination and a 24-h Holter ECG monitoring. 

### 3.6. Comparison between Childbearing and Nulliparous Subjects

Genetic analysis identified a P/LP genetic variant in 11 (69%) childbearing women and in 15 (79%) nulliparous women (*p* = 0.490); however, this was without significant differences regarding the prevalence of specific disease genes in the two groups. The age of nulliparous women was significantly lower when compared to women with previous pregnancies (Table 1). There were no differences between the two groups regarding the ratio of probands, the family history of cardiomyopathies, HF, or ECG features. Regarding morphological abnormalities, a two-D echocardiogram comparison of RV dimensions was not significant (*p* = 0.060) (see Table 2). CMR assessment showed no differences between the two groups regarding LV and RV dimensions and function; however, WMA and the presence of LGE were reported. The 24-h ECG Holter evaluation of 32 patients (91%) showed VAs (isolated PVBs in 20 patients, 57%, and NSVT in 12 patients, 34%) without significant differences between the nulliparous and childbearing women.

### 3.7. Follow-Up Analysis

The follow-up period had a mean duration in the overall cohort of 7.80 ± 6.69 years (min 1–max 30 years). In detail, the mean duration for childbearing women was 8.31 ± 6.07 (min 2–max 20 years), while for nulliparous women it was 8.31 ± 7.37 (min 1–max 30 years), which demonstrated a *p* = 0.684. Considering overall events, seven patients (20%) experienced an sVT. Of these, four (57%) belonged to the childbearing group and all episodes occurred after the last pregnancy, with a median time interval of 11.5 years (min 2 months–max 27 years). Furthermore, the age at sVT onset did not differ significantly between the groups of nulliparous (median 33, min 29, max 36 years) and childbearing women (median 45, min 33, max 59 years), which demonstrated a *p* = 0.100. In a total of ten patients (29%) an ICD was implanted (age at implant 36 ± 7 years, min 25, max 45 years), whereas five (50%) patients possessed primary prevention (3 belonging to childbearing and 2 to nulliparous group). Two patients, one belonging to each group, had HF episodes at the age of 42 and 38 years, respectively. Kaplan–Meir analysis of the VT-free survival curves showed no significant difference between the groups of pregnant and nulliparous women (Log Rank *p* = 0.220). Furthermore, no association was found between pregnancy and MVA (HR 0.77, C.I. 95% 0.17–3.46, and *p* = 0.728).

## 4. Discussion

ALVC is a recently described clinical entity in which clinical manifestations, effective therapy, outcome, and risk stratification are not completely understood [5]. To the best of our knowledge, this is the first study that describes pregnancy course and outcome in a cohort of female patients that are diagnosed with this disease. Furthermore, our purpose was to characterize the effect of pregnancy on this type of cardiomyopathy through the comparison of a series of ALVC women who underwent a pregnancy with a group of affected nulliparous patients. We found that pregnancy is well tolerated in ALVC patients, and that history of pregnancy does not seem to modify the outcome in terms of either electrical instability or HF.

### 4.1. Physiological Cardiac Changes in Pregnancy

Different physiological changes in the cardiovascular system occur during pregnancy. The maternal blood volume increases significantly starting at around 6 weeks of gestation and reaches a maximal volume by the 32 weeks period with a comprehensive increase of 45% [21]. Furthermore, there is a comparable increase in cardiac output, which is achieved by a rise in stroke volume, and later in pregnancy with an increase in heart rate. The growth in plasma volume and cardiac output are triggered by vasodilation, which occurs early in pregnancy due to hormonal influences. The above circulatory changes are matched by an increase in the LV wall muscle and EDV, with end-systolic volume and end-diastolic pressure that remain unchanged [21]. Overall, LV systolic function improves early in pregnancy and progresses gradually until 20 weeks’ gestation due to LV afterload reduction. Hemodynamic changes are fully reset after 6 months [22].

### 4.2. Pregnancy in Patients with ACM

Available data seem to indicate that in the majority of women with ACM the course of pregnancy and postpartum are uneventful with regard to pregnancy-related mortality and complications [23]. Depending on the number of patients enrolled in the study, individual risk profile, and duration of follow-up, the rates of maternal death from all causes varied from 0 to 4%. Moreover, the comparison of ACM patients who experienced a pregnancy with nulliparous affected patients demonstrated no difference in clinical event rates and acceleration of VA and HF, either during pregnancy or after childbirth, both early and long after [9,24,25]. Thus, published data appears to prove that pregnancy does not constitute a driving force for disease progression in ACM [23]. Furthermore, the number of pregnancies appear to have no impact on the outcome and incidence of maternal complications [9].

### 4.3. Pregnancy in Patients with DCM

DCM is characterized by dilation and impaired contraction, primarily of the left ventricle (LV). ALVC may overlap with DCM when LV dilation and systolic dysfunction are present in both conditions. Further, CMR assessment has an important role in providing a differential diagnosis of the extent of LV-LGE. Moreover, this predominantly affects the inferolateral segments and is, also, significantly greater in ALVC [3]. Only a few studies on the pregnancy outcomes of women with idiopathic DCM have been published so far [26,27,28,29]. Having said that, pregnancy in asymptomatic or mildly symptomatic DCM patients appears to be associated with a low risk of adverse maternal events. However, pregnancy is poorly tolerated in some women with pre-existing DCM, with the potential for significant deterioration in LV function. In addition, the predictors of maternal mortality are found in the NYHA class III/IV as well as at EF <40%. Highly adverse risk factors include EF <20%, the presence of mitral regurgitation, as well as RV failure and/or hypotension [30]. Furthermore, pregnant women with DCM experience more adverse events compared to non-pregnant women [28,31], and this could be partially explained by the increased hemodynamic challenge that occurs in pregnant patients and by the need to discontinue ACE-inhibitors during pregnancy due to their teratogenic effects.

### 4.4. Pregnancy Outcome in Patients with ALVC

In our patients, pregnancies were well tolerated, and none experienced adverse events. Nonetheless, it is important to underline that in our series LV systolic function was found to be severely reduced before pregnancy (EF < 35%) in only 1 of the 16 women belonging to the childbearing group, thus further data on a larger cohort are required for more comprehensive knowledge. One patient showed an sVT episode two months after the third pregnancy, however the same arrhythmic events also occurred two years later, thus a clear relationship between sVT and pregnancy cannot be proved. Furthermore, pregnancies do not seem to have a role on the degree of electrical instability, considering that the number of patients with sVT, as well as age of patients at the time of arrhythmic episodes, did not differ significantly in the two groups.

### 4.5. Pregnancy Management in ALVC Patients

As in other cardiomyopathies, women with ALVC who wish to have a baby require a pre-pregnancy risk assessment, counselling, and should be reviewed by a pregnancy heart team with a cardiologist and a gynecologist. Considering the lack of data on pregnancy tolerance in ALVC patients, we should also consider the existing data on DCM, which demonstrate that the NYHA class III/IV and EF <40% are predictors of maternal mortality, with highly adverse risk factors when EF <20%. Pre-pregnancy management should include modification of the existing medications in order to avoid teratogenicity and to minimize harm to the fetus. Even if the appropriate management strategy for ALVC is not completely established, in the presence of LV dilation/dysfunction ACE-inhibitors and angiotensin receptor blockers are frequently prescribed, as well as ARNI or mineralocorticoid receptor inhibitors; these are contraindicated during pregnancy and should be discontinued prior to conception, with close clinical and echocardiographic monitoring. Beta-adrenergic blocking agents are generally safe in pregnancy even if they are associated with increased rates of fetal growth restriction [30]. Flecainide has been safely used to treat maternal and fetal arrhythmias; however, although rare, neonatal toxicity can occur [31]. Patients should be monitored during pregnancy and postpartum with periodic cardiac evaluation with ECG assessments, 2-D-echocardiogram tests, and 24-h Holter ECG evaluations, and close cooperation with gynecologists. Additionally, if indicated, ICD implant is safe during pregnancy [32,33].

### 4.6. Possible Role of Pregnancy in Disease Progression in ALVC Patients

As expected, at the time of evaluation, patients with previous pregnancies were significantly older than nulliparous ones. Thus, considering that we are dealing with a progressive myocardial disease, the specific role of pregnancies in disease progression is difficult to estimate. Nonetheless, despite a significant age difference, the two groups of patients did not differ significantly regarding LV dimensions and function, as well as in the degree of electrical instability, thus suggesting the absence of a clear role of pregnancy in disease progression.

## 5. Conclusions

Pregnancy in ALVC patients with normal or mildly reduced systolic function, and without previous HF episodes, appear to be well tolerated and do not appear to have a role in disease progression, either in terms of ventricular dilatation/dysfunction or in the degree of electrical instability. Further studies on a larger cohort of patients with different degrees of disease extent and genetic background are needed in order to achieve a more comprehensive knowledge on the outcome of pregnancy in this disease.

## Figures and Tables

**Figure 1 jcm-11-06735-f001:**
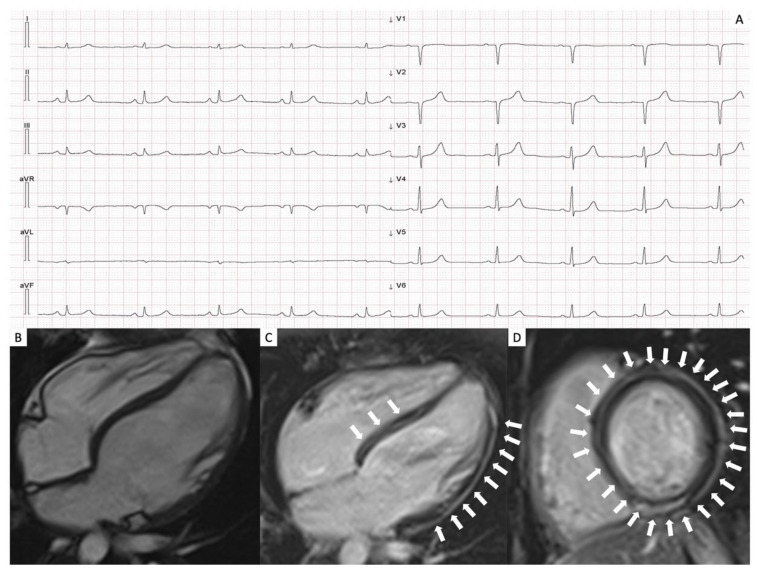
ECG and CMR assessments of a childbearing patient carrying a likely pathogenic variant of PKP2. The ECG assessment showed the presence of low QRS voltages (panel (**A**)). CMR evaluation demonstrated normal size and biventricular function (panel (**B**)), as well as extensive signs of fibrous infiltration with sub-epicardial “ring-like” distribution in LGE sequences (panels (**C**,**D**)). ECG: electrocardiogram; CMR: cardiac magnetic resonance; and LGE: late gadolinium enhancement.

**Table 1 jcm-11-06735-t001:** Clinical and ECG features of the 35 ALVC female patients.

	Overall *n* = 35	Childbearing Women *n* = 16	Nulliparous Women *n* = 19	*p*
Probands	18 (51%)	8 (50%)	10 (53%)	0.573
Age at evaluation	40 ± 11	45 ± 9	36 ± 11	0.050
Family history of SCD	18 (51%)	7 (44%)	11 (58%)	0.505
Family history of DCM	9 (26%)	5 (31%)	4 (21%)	0.700
Family history of ALVC	18 (51%)	8 (50%)	10 (52%)	1.000
Family history of ARVC	11 (31%)	5 (31%)	6 (32%)	1.000
Family history of HF	11 (31%)	8 (50%)	3 (16%)	0.065
Myocarditis-like episodes	4 (11%)	1 (6%)	3 (16%)	0.608
Ventricular arrhythmias	32 (91%)	15 (94%)	17 (89%)	1.000
Frequent PVBs	13 (37%)	4 (25%)	9 (42%)	0.311
NSVT	12(41%)	7 (50%)	5 (32%)	0.468
Sustained VT	7 (20%)	3 (19%)	4 (21%)	0.799
ICD	10 (29%)	5 (31%)	5 (32%)	0.747
Normal ECG	10 (28%)	4 (25%)	6 (31%)	0.099
Low QRS voltages (limb leads)	23 (65%)	13 (81%)	10 (53%)	0.152
Low QRS voltages (precordial leads)	11 (31%)	5 (31%)	6 (32%)	1.000
Negative T wave V1-V3	4 (11%)	1 (6%)	3 (16%)	0.608
Negative T wave V4-V6	7 (20%)	3 (19%)	4 (21%)	1.000
Negative T wave inferior leads	6 (17%)	2 (12%)	4 (21%)	0.666

SCD: sudden cardiac death; DCM: dilated cardiomyopathy; ALVC: arrhythmogenic left ventricular cardiomyopathy; ACM: arrhythmogenic right ventricular cardiomyopathy; HF: heart failure; PVBs: premature ventricular beats; VT: ventricular tachycardia; NSVT: non-sustained VT; and ICD: implantable cardioverter–defibrillator.

**Table 2 jcm-11-06735-t002:** ACM genetic variants identified in 29 women affected with ALVC (see text).

Pt	Cohort	Gene	cDNA Change	Amino Acid Change	ACMG Classification	Reference
#1	*NW*	*DSP*	c.3465G > A	p.Trp1155*	P (PVS1, PM2, PP5)	
#2 and #3	*NW, CW*	*DSP*	c.939 + 1G > A	/	P (PVS1, PM2, PP5)	Whittock et al., 1999 [15]
#4	*NW*	*DSP*	c.3475G > T	p.Glu1159*	P (PVS1, PM2, PP5)	Bariani et al., 2021 [16]
#5	*NW*	*DSP*	c.132delG	p.Arg45Alafs*3	LP (PVS1, PM2)	
#6	*NW*	*DSP*	c.897C > G	p.Ser299Arg	LP (PM2, PP3, PP5, PS3)	Rampazzo et al., 2002 [17]
#7	*NW*	*DSP*	c.2821C > T	p.Arg941*	P (PVS1, PM2, PP5)	Quarta et al., 2011 [18]
#8	*CW*	*DSP*	c.939 + 1G > A	/	P (PVS1, PM2, PP5)	Whittock et al., 1999 [15]
#9 and #10	*NW, NW*	*DSP*	c.3891_3894dupGGTC	p.Met1299Glyfs*7	P (PVS1, PM2, PP5)	Bariani et al., 2022 [14]
#11	*NW*	*DSP*	c.337C > T	p.Gln113*	P (PVS1, PM2, PP5)	Bariani et al., 2022 [14]
#12	*NW*	*DSP*	c.2297 + 1G > T	/	P (PVS1, PM2, PP5)	Bariani et al., 2022 [14]
#13	*NW*	*DSP*	c.974_975delAG	p.Glu325Alafs*3	P (PVS1, PM2, PP5)	Bariani et al., 2022 [14]
#14 and #15	*CW, CW*	*DSP*	c.3889C > T	p.Gln1297*	P (PVS1, PM2, PP5)	Bariani et al., 2021 [14]
#16	*CW*	*DSP*	c.6850C > T	p.Arg2284*	P (PVS1, PM2, PP5)	Fressart V et al., 2010 [19]
#17 and #18	*CW, CW*	*DSP*	c.3416dupA	p.Tyr1139*	P (PVS1, PM2, PP5)	Bariani et al., 2022 [14]
#19	*CW*	*DSP*	c.423-1G > A	/	P (PVS1, PM2, PP5)	Bariani et al., 2022 [14]
#20	*CW*	*DSP*	c.4207_4208delAG	p.Arg1403Glufs*4	LP (PVS1, PM2)	
#21	*NW*	*DSP*	c.1067C > T	p.Thr356Met	VUS (PM2, PP3)	
#22	*NW*	*DSG2*	c.3059_3062delAGAG	p.Glu1020Alafs*18	VUS (PVS1, BS2)	Christensen et al., 2010 [20]
#23	*NW*	*FLNC*	c.5926C > T	p.Gln1976*	P (PVS1, PM2, PP5)	Celeghin et al., 2021 [13]
#24 and #25	*NW, CW*	*FLNC*	c.5398 + 1G > T	/	P (PVS1, PM2)	Celeghin et al., 2021 [13]
#26	*NW*	*FLNC*	c.376_392delAACCTGAAGCTGATGCT	p.Asn126Glyfs*20	P (PVS1, PM2)	Celeghin et al., 2021 [13]
#27	*CW*	*FLNC*	c.7037dup	p.Leu2347Profs*9	P (PVS1, PM2, PP5)	Celeghin et al., 2021 [13]
#28	*NW*	*PKP2*	c.1521G > A	p.Trp507*	LP (PVS1, PM2)	
#29	*CW*	*PKP2*	c.2443_2448delAACACCinsGAAA	p.Asn815Glufs*11	P (PVS1, PM2, PP5)	

ACMG: American College of Medical Genetics and Genomics; P: pathogenic; LP; likely pathogenic; DSP: desmoplakin; PKP2: plakofillin-2; DSG2: desmoglein-2; FLNC: filamin-C; CW: childbearing woman; and NW: nulliparous woman.

**Table 3 jcm-11-06735-t003:** Imaging findings of the 35 ALVC female patients.

	Overall *n* = 35	Childbearing Women *n* = 16	Nulliparous Women *n* = 19	*p*
**Echocardiographic findings**				
LVEDVi (mL/m^2^)	76 ± 15	80 ± 15	73 ± 15	0.104
LVESVi (mL/m^2^)	39 ± 13	42 ± 12	36 ± 12	0.088
LVEF (%)	50 ± 7	48 ± 7	52 ± 7	0.230
LV kinetic abnormalities	17 (49%)	8 (50%)	9 (47%)	0.370
RVA (cm^2^)	15,5 ± 3	16 ± 3	15 ± 3	0.092
RVAC (%)	40 ± 5	39 ± 6	41 ± 4	0.147
**CMR findings**	***n* = 32**	***n* = 13**	***n* = 19**	
LVEDVi (mL/m^2^)	89 ± 17	93 ± 19	88 ± 14	0.377
LVESVi (mL/m^2^)	53 ± 8	45 ± 16	42 ± 6	0.734
LVEF (%)	52 ± 8	53 ± 9,8	53 ± 7	0.827
LV WMA	17 (49%)	10 (63%)	7 (37%)	0.181
CMR RVEDVi (mL/m^2^)	77 ± 15	74 ± 14	79 ± 16	0.472
CMR RVEF (%)	60 ± 9	55 ± 6	58 ± 11	0.384
WMA RV	13 (37%)	6 (46%)	7 (37%)	0.720
FAT LV	12 (38%)	5 (39%)	7 (37%)	1.000
LGE LV	32 (100%)	13 (100%)	19 (100%)	1.000
LGE LV > 2 segments	25 (78%)	10 (77%)	15 (79%)	1.000
FAT RV	5 (16%)	4 (31%)	1 (5%)	0.132
LGE RV	2 (6%)	1 (8%)	1 (5%)	0.780
LGE RV > 2 segments	1 (3%)	1 (8%)	0 (0%)	0.406

CMR: cardiac magnetic resonance; LV: left ventricular; RV: right ventricular; LVEDVi: indexed LV end-diastolic volume; LVESVi: indexed LV end-systolic volume; LVEF: LV ejection fraction; RVA: RV area; RVAC: RV area change; WMA: wall motion abnormalities; and LGE: late enhancement.

## Data Availability

The data presented in this study are available on request from the corresponding author.

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
