# Peer review of "Pregnancy in Women with Arrhythmogenic Left Ventricular Cardiomyopathy"

_jcm, 2022, doi:10.3390/jcm11226735_

Round 1
Reviewer 1 Report
Table 2. unlike other tables, for some reasons, this table does not include category of childbearing group and nulliparous group. what group do these genetic variants belong to (childbearing group or nulliparous group)?
Table3. Is the unit(ml/m2 ) correct for LVEDV or RVEDV? ml/m2 is the unit for end-diastolic volume index (EDVI). Because the end-systolic volume is an essential parameter used for the assessment of cardiac function, do you have data of LVESV(end systolic value)? In the table 3, you had the comparison between childbearing women and nulliparous women. In the manuscript, however, you wrote that “ LV end diastolic volume (EDV) was increased in 24 patients (89%) and LV ejection fraction (LV-EF) was reduced in 18 (51%), with the presence of regional kinetic abnormalities in 15 (43%).” Can you better clarify this?
You simply pointed to Figure 1 in the manuscript, can you better describe each panel of figure 1 in the manuscript?
Panel A of figure2: why the survival curve in the graph did not drop down when the number of survivors already become significantly less? For example, at 48 or 60 years, nulliparous women dropped to 2 and 1, and childbearing women dropped to 0 at 60-year-old, but the survival curve still showed above 50% survival rate.
Author Response
- Table 2. unlike other tables, for some reasons, this table does not include category of childbearing group and nulliparous group. what group do these genetic variants belong to (childbearing group or nulliparous group)?
Thank you. As suggested by the Reviewer, we specified the cohort of patients (Nulliparous woman-NW and childbearing woman-CW, respectively)
- Is the unit (ml/m2) correct for LVEDV or RVEDV? ml/m2 is the unit for end-diastolic volume index (EDVI). Because the end-systolic volume is an essential parameter used for the assessment of cardiac function, do you have data of LVESV (end systolic value)?
In the table 3, you had the comparison between childbearing women and nulliparous women. In the manuscript, however, you wrote that “LV end diastolic volume (EDV) was increased in 24 patients (89%) and LV ejection fraction (LV-EF) was reduced in 18 (51%), with the presence of regional kinetic abnormalities in 15 (43%).” Can you better clarify this?
Thank you for your comment. The unit of measure is correct, all values are indexed. We have made changes in Table 3. To better clarify the results, we modified the text (page 8, lines 198-199).
- You simply pointed to Figure 1 in the manuscript, can you better describe each panel of figure 1 in the manuscript?
Thank you for the suggestion. The figure was described in the text, p. 7 lines 182-183, p. 8 lines 205-206.
- Panel A of figure2: why the survival curve in the graph did not drop down when the number of survivors already become significantly less? For example, at 48 or 60 years, nulliparous women dropped to 2 and 1, and childbearing women dropped to 0 at 60-year-old, but the survival curve still showed above 50% survival rate.
Thank you for your comment. This fact could be explained considering that the number decreases as it is "censored" and not due to an event, so the curve does not drop.
Reviewer 2 Report
The manuscript with interesting topic. Hoever, some queations need answer.
1.Please explain why the LVEF of Echo and CMR between Childbearing and Nulliparous groups are different? the LVEF of Echo in Childbearing women is lower than Nulliparous women, but that of CMR in Childbearing women is higher than Nulliparous women.
2.Why the Kaplan Mayer curves showed the tendency to separate, but no significant difference between the groups of pregnant and nulliparous women,is it related with the small sample size or time of follow-up? I think this is a limitation of this study and should be discussed in the conclusion.
3.The figure of Kaplan Mayer curves is not very clear.
Author Response
- Please explain why the LVEF of Echo and CMR between Childbearing and Nulliparous groups are different? the LVEF of Echo in Childbearing women is lower than Nulliparous women, but that of CMR in Childbearing women is higher than Nulliparous women.
Thank you for the comment. The observed difference could be explained by the different methods used. CMR tends to provide volumes higher values compared to echocardiography, as pointed out by several publications.
- Why the Kaplan Mayer curves showed the tendency to separate, but no significant difference between the groups of pregnant and nulliparous women,is it related with the small sample size or time of follow-up? I think this is a limitation of this study and should be discussed in the conclusion.
- The figure of Kaplan Mayer curves is not very clear.
Thank you for the comments. The reason of this fact could be the low sample size. A comment about the need to increase the sample size is already included in the conclusion chapter, p. 12 lines 357-359.
Since the figure does not add additional information compared to data already reported in the text, we decided to remove Figure 2.
Reviewer 3 Report
Thank you for the opportunity to review this article describing pregnancy in women with ALVC.
The manuscript describes a data collection process in consecutive female ALVC patients in the cardiomyopathy clinic at Padua University Hospital. However, there is no description of the following:
- Data collection period/timeframe. Over what period did these women attend? What was the mean follow-up period for each participant
- Inclusion/exclusion criteria for this study. How many, if any, were excluded and what percentage of the total number of Padua female ALVC patients does this cohort represent?
- No description of the medical regime of these patients.
- No description of what monitoring took place during pregnancy or what biomarkers where measured before/during or after pregnancy.
Table 2 does not add any value to this article.
Although this manuscript concludes that pregnancy is well tolerated, it does not mention that four patients (25% of the ALVC pregnancy cohort) underwent necessary c-section; two for ALVC related reasons and two for foetal growth delay. You also note that Beta-adrenergic blocking is associated with foetal growth delay but have not discussed this in the context of your findings - the reason for 2 c-section. Although C-section is not an adverse event, that it was required in 25% of your cohort because of ALVC and it's treatment is an important finding.
The relevance of a paragraph on pregnancy in DCM (section 4.3) is unclear in an ALVC specific manuscript.
Rates vary per country, but the rate of maternal death in the UK is around 9 per 100,000 births. However, although this manuscript quotes in section 4.2 a maternal mortality rate of 0-4% (up to 1 death per 25 births), it concludes that pregnancy in ALVC is well tolerated. This needs clarification.
There is no description of whether these patients (childbearing and nulliparous) underwent careful clinical review and counselling prior to becoming pregnant. If standard practice at Padua is to advise symptomatic AVC patients not to become pregnant, this small cohort of 16 patients may represent those at lowest risk of VA or SCD during pregnancy. This is not discussed and has important implications.
Although I appreciate that the prevalence of ALVC is low and therefore limits the potential cohort size, there are just too few participants in this study for the claims of this manuscript to be confirmed - this is reflected in the level statistical significance reached.
You state that pregnancy does not have a role in disease progression but have not described anywhere the follow-up period, nor have you reported pre and post pregnancy testing to confirm that ALVC disease has not progressed. Is the suggestion of stable disease based upon symptom reporting and documented VT? If so, this does not confirm your statement
Author Response
Thank you for the opportunity to review this article describing pregnancy in women with ALVC.
The manuscript describes a data collection process in consecutive female ALVC patients in the cardiomyopathy clinic at Padua University Hospital. However, there is no description of the following:
- Data collection period/timeframe. Over what period did these women attend? What was the mean follow-up period for each participant
Thank you for your comment. The duration of the recruitment period has been added on page 2, line 77. Moreover, the follow-up period had a mean duration in the overall cohort of 7.80±6.69 years (min 1 – max 30 years). In details, mean duration for childbearing women was 8.31±6.07 (min 2-max 20 years), while for nulliparous women was 8.31±7.37 (min 1-max 30 years), p=0.684. The data were added in the text on page 9, lines 241-243.
- Inclusion/exclusion criteria for this study. How many, if any, were excluded and what percentage of the total number of Padua female ALVC patients does this cohort represent?
Thank you for your comment. In the study we included all women with ALVC evaluated in Padua in the period 1990-2020.
- No description of the medical regime of these patients.
The majority of patients were diagnosed with ALVC after pregnancy. Regarding medical therapy, after diagnosis 14 (88%) used beta-blockers (mainly metoprolol), in 7 cases (50%) associated with ACE inhibitor, 1 amiodarone (6%), 1 sotalol associated with ACE inhibitor (6%). In the patients diagnosed before pregnancy, therapy was modified by discontinuing the ACE inhibitor, while beta-blocker therapy was maintained. Data were added to the text on page 5, lines 156-163.
- No description of what monitoring took place during pregnancy or what biomarkers were measured before/during or after pregnancy.
Patients diagnosed with ALVC before pregnancy underwent a close cardiac and obstetrical monitoring. From the cardiological point of view, each patient underwent a cardiological examination including an echocardiographic examination and 24-hour Holter ECG monitoring. Data were added to the text on page 9, lines 224-227.
- Table 2 does not add any value to this article.
We agree that the role of genetic background in clinical management of pregnancy in ALVC patients has not been studied so far; nonetheless, we believe that as genetic data are a fundamental part of ALVC diagnosis, it could be important to report this information.
- Although this manuscript concludes that pregnancy is well tolerated, it does not mention that four patients (25% of the ALVC pregnancy cohort) underwent necessary c-section; two for ALVC related reasons and two for fetal growth delay. You also note that Beta-adrenergic blocking is associated with fetal growth delay but have not discussed this in the context of your findings - the reason for 2 c-section. Although C-section is not an adverse event, that it was required in 25% of your cohort because of ALVC and its treatment is an important finding.
Thank you for your suggestions. As correctly stated by the Reviewer, beta-blockers have been proved to be a cause of fetal grow delay and both our patients in whom a fetal group delay was demonstrated were on beta-blockers therapy, actually. This information was added in the Results (page 9, lines 222-223). Regarding the c-sections due to ALVC related reasons, they were performed in one patient with a severely decreased LV systolic function (EF: 35%) and in another one with a previous syncope and several episodes of NSVT.
- The relevance of a paragraph on pregnancy in DCM (section 4.3) is unclear in an ALVC specific manuscript.
We introduced the paragraph on DCM as, considering the lack of data on pregnancy tolerance in ALVC patients, we thought that existing data on DCM should be useful, as this disease is characterized by dilation and impaired contraction, primarily of the left ventricle (LV), similarly to ALVC forms.
- Rates vary per country, but the rate of maternal death in the UK is around 9 per 100,000 births. However, although this manuscript quotes in section 4.2 a maternal mortality rate of 0-4% (up to 1 death per 25 births), it concludes that pregnancy in ALVC is well tolerated. This needs clarification.
Thank you for your comment. The reported maternal mortality rates on ACM pregnant women derives from studies on patients with ACM with all different phenotypes, even if the “classical” and biventricular forms are the most represented. Indeed, our small cohort represents the first available case series of childbearing ALVC women and the small size of the cohort does not allow to estimate the maternal mortality rate in this subgroup of patients.
- There is no description of whether these patients (childbearing and nulliparous) underwent careful clinical review and counselling prior to becoming pregnant. If standard practice at Padua is to advise symptomatic AVC patients not to become pregnant, this small cohort of 16 patients may represent those at lowest risk of VA or SCD during pregnancy. This is not discussed and has important implications.
Thank you for your comment. The topic was discussed in section 4.5. All ACM patients referred to the Padua cardiomyopathy center are usually informed about the risks of pregnancy and counselling is always carried out. Similarly, it was discussed in section 4.4. that all patients enrolled in the study had all but one EF > 35%, thus further data on a larger cohort are needed for a more comprehensive knowledge.
- Although I appreciate that the prevalence of ALVC is low and therefore limits the potential cohort size, there are just too few participants in this study for the claims of this manuscript to be confirmed - this is reflected in the level statistical significance reached.
Thank you. As already pointed out in the conclusions, the low sample size is a limitation of the study and new multicenter studies with larger population sizes are needed.
- You state that pregnancy does not have a role in disease progression but have not described anywhere the follow-up period, nor have you reported pre and post pregnancy testing to confirm that ALVC disease has not progressed. Is the suggestion of stable disease based upon symptom reporting and documented VT? If so, this does not confirm your statement.
Thank you for your comment. As ALVC is a progressive disease, it is difficult to estimate the impact of pregnancy on the evolution of the disease. In our study, in ALVC women with LVEF>35% and NYHA class I^ pregnancy was well tolerated. Moreover, no clinical and instrumental differences between nulliparous and childbearing groups were reported. Furthermore, our data did not show significant differences between the two groups regarding the free survival at follow-up (Log-Rank p=0.220) and the incidence of major arrhythmic events (HR 0.77, C.I. 95% 0.17-3.46, p=0.728).
Round 2
Reviewer 1 Report
The authors well address the points I raised. They also revise and clearly improve the quality of manuscript. The figures/tables are appropriate and properly show the data. The conclusion is consistent with the data.